# Understanding the impact of delegated home visiting services accessed via general practice by community-dwelling patients: a realist review protocol

Ruth Abrams,[1] Geoffrey Wong,[2] Kamal Ram Mahtani,[2] Stephanie Tierney,[2] Anne-Marie Boylan,[2] Nia Roberts,[3] Sophie Park[1]

[1]Department of Primary Care and Population Health, Institute of Epidemiology and Public Health UCL Medical School, London, UK
[2]Nuffield Department of Primary Care Health Sciences, University of Oxford, Oxford, UK
[3]Bodleian Health Care Libraries, University of Oxford, Oxford, UK

**Correspondence to**
Mrs Ruth Abrams;
r.abrams@ucl.ac.uk

## ABSTRACT

**Introduction** In western countries, early visiting services (EVS) have been proposed as a recent intervention to reduce both general practitioner workload and hospital admissions among housebound individuals experiencing a healthcare need within the community. EVS involves the delegation of the patient home visits to other staff groups such as paramedics or nursing staff. However, the principles of organising this care are unknown and it remains unclear how different contexts, such as patient conditions and the processes of organising EVS influence care outcomes. A review has been designed to understand how EVS are enacted and, specifically, who benefits, why, how and when in order to provide further insight into the design and delivery of EVS.

**Methods and analysis** The purpose of this review is to produce findings that provide explanations of how and why EVS contexts influence their associated outcomes. Evidence on EVS will be consolidated through realist review—a theory-driven approach to evidence synthesis. A realist approach is needed as EVS is a complex intervention. What EVS achieve is likely to vary for different individuals and contexts. We expect to synthesise a range of relevant data such as qualitative, quantitative and mixed-method research in the following stages: devising an initial programme theory, searching evidence, selecting appropriate documents, extracting data, synthesising and refining the programme theory.

**Ethics and dissemination** A formal ethics review is not required as this study is secondary research. Findings will be disseminated in a peer-reviewed journal, at national and international conferences and to relevant professional associations.

**PROSPERO registration number** CRD42018096518.

## Strengths and limitations of this study

► To our knowledge, this is the first realist review to synthesise evidence and produce subsequent conceptualisations on the use and implementation of early visiting services (EVS).
► Undertaking a realist review enables us to understand the complexity of EVS and accounts for the different outcomes they cause under varying contexts, and makes our work potentially transferable.
► Stakeholder engagement during programme theory development accounts for a range of perspectives aiding this review's relevance for other professionals.
► Studies in only English language will be included.

also a financial challenge to sustaining current healthcare practices, particularly in light of an ageing population with increasingly complex health conditions.[2] In particular, housebound individuals may find it challenging to access primary care at general practice surgeries and have had to access emergency departments for alternative care solutions.[3–5] This places a strain on emergency departments and may lead to an increased likelihood of hospitalisation, particularly in older adults, which may be detrimental to their overall well-being in the longer term.[6]

In addition to increased patient demand, difficulties in general practice recruitment and retention means that general practitioners (GPs) are currently facing a number of hurdles in terms of workload.[7] As such, the opportunity for GPs to provide visits to housebound patients is challenging and data indicate that GP home visit rates are decreasing in many European and Northern America countries.[8 9]

One possible way to reduce the pressures GPs face and provide access (especially for housebound patients) to care, is to delegate patient home visits to other

## INTRODUCTION
### Background

Current practices within the National Health Service (NHS) aim to reduce unnecessary hospital admissions because (1) current bed occupancy rates for general and acute settings are high, (2) emergency attendances and emergency admissions are increasing and (3) patient waiting times are increasing.[1] There is

staff groups such as paramedics, physician assistants or nursing staff.[10 11] Delegating GP home visits to other staff groups may improve the accessibility and timeliness of care received.[12] As such, early visiting services (EVS) has been proposed and piloted as a possible intervention.[13]

### The nature of EVS

Whether a GP decides to delegate a visit may depend on a number of factors. For example, patients may present with a clear cut acute need that is untreatable at home and inevitably requires a hospital admission (eg, in the case of a myocardial infarction or stroke). GP workload, practice location and GP tolerance of patient discomfort are also factors that affect the likelihood of a home visit being delegated.[9]

GPs are more likely to delegate a visit if they perceive it will save them time and contribute to patient health.[10] Positive GP perceptions about the delegation of home visits to other staff members have been reported as acceptable in a number of studies. For example, staff perceived as acceptable include nurses,[11 14] emergency care practitioners[15] and physician assistants.[10]

Yet, EVS may vary in its purpose and delivery. Our initial scoping and discussions with stakeholders have identified a variety of ways in which an EVS is described and used. As such, this intervention may come at different points in a patient's care pathway ranging from preventative to acute, depending on patient needs.[16]

### EVS: a preventative measure

Some define EVS as an 'early' preventative measure to provide routine care in a patient's home and minimise subsequent deterioration. Typically, this includes supporting individuals at risk from a chronic condition such as chronic obstructive pulmonary disease, diabetes, dementia or frailty. Patients with chronic conditions may be offered interventions such as geriatric assessments, falls prevention, dietary intervention and/or medication review.[17]

These have all been shown to have a positive effect on the assessment and management of physical functioning; psychosocial functioning; falls; hospital admission and mortality, particularly among the elderly.[18 19] However, extant research into the effectiveness of preventative home visits within primary care has frequently been cited as inconclusive.[17–19] The associated benefits of home visits and their ability to offer anything conclusive are affected by the differences in intervention components and delivery methods.[20] Moreover, randomised control trials have shaped the way in which home visit effectiveness has been examined. This has led to a neglected understanding of their justification and benefit,[21] particularly in relation to contextual factors (such as economic status), patient psychosocial factors (such as, eg, social networks) and restricted outcome measures (eg, mortality rates or function with a specific population).

### EVS: an 'earlier in the day' intervention

The second way in which the term EVS has been used is as a responsive intervention to ensure patients have access to care earlier in the day. This assumes that by having patients seen earlier in the day, EVS visits act either as preventative of hospital admissions and overnight stays or enable a patient to be admitted earlier in the day, spreading out the flow of work at hospitals. When delegated, GP workload is also thought to be reduced by removing the need to fit in a home visit.[22] Therefore, greater efficiency in processing a patient through the healthcare system is emphasised.

The organisation of GP workload, however, is likely to vary across practices and inevitably impact on when a home visit might be undertaken. Some practices may have access to a duty doctor throughout the day or position home visits after morning surgery. Moreover, relatively few studies have examined access to GP services and their associated hospital admission route.[3]

### EVS: an acute same day service

The third way in which EVS has been described is for patients with an acute, same day need.[13] An acute need can be defined as a condition with a finite duration[23] such as acute injury, acute exacerbation of chronic disease and acute minor illness that prevents them from accessing traditional GP services.[11] However, the process of decision-making about a patient's care in an acute context is complex, with professional and patient thresholds of risk likely to be variable[24] or reliant on the medical autonomy of the qualified professional treating the patient.[25]

This different conceptualisation indicates that our current understanding of EVS is poor. Descriptions of the purpose and way EVS are provided differ and at present it is unclear what outcomes EVS might achieve, how, why, for whom and in what contexts. Thus, consolidation of evidence regarding EVS is now required.

## METHODS
### Review aim, questions and objectives
#### Aim
This review aims to improve our understanding of the ways in which (ie, how, why and in what contexts) EVS impact (or not) on hospital admissions, GP workload and patient health within primary care settings.

Review objectives:
1. To conduct a realist review to understand the ways in which EVS impact on the healthcare needs of community-dwelling patients. This will be done with (A) engagement with a diverse range of literature, (B) the development of a programme theory and (C) feedback and advice from stakeholders experienced in the field.
2. To produce recommendations that guide the implementation and commissioning of EVS within primary care.

Review research questions:

Within the existing and available literature, what are the causal explanations for the ways in which primary care EVS contribute to patient care and clinical workload?

## Sub questions

1. What are the outcomes from EVS?
2. What are the mechanisms, acting at individual, group, professional and/or organisational levels, through which EVS result in their outcomes?
3. What are the contexts which determine whether the different mechanisms produce their outcomes?

## STUDY DESIGN

Our review design is based on the work of Pawson *et al*[26] and the project protocol by Carrieri *et al*[27] and Weetman *et al*.[28] It takes a realist approach, viewing causation as a generative process—where outcomes are caused by context sensitive mechanisms.[26] We have conceptualised EVS as a complex intervention that has outcomes which are context sensitive. Therefore, our review approach will enable us to identify and understand the contexts in which the outcomes of EVS may or may not be effective.

A realist review is able to synthesise a range of relevant data such as qualitative, quantitative and mixed-method research, as well as grey literature. Realist reviews move beyond a description of literature by using an interpretive, theory-driven approach to analysing data from such diverse literature sources. Findings from our realist review are potentially transferable because we will focus on the mechanisms that cause particular EVS outcomes. This may enable us to produce recommendations likely to be useful across the NHS and possibly further afield.

## Patient and public involvement

The realist review protocol incorporates iterative cycles of engagement with the literature and with our Stakeholder Group. Our stakeholders comprise a group of individuals involved in the undertaking or organisation of EVS including Clinical Commissioning Group members, emergency care practitioners and GPs. These individuals were identified from internet searches of general practices running EVS, or professional networks of the authors and invited to have an informal conversation about the operationalisation of EVS at their practice. Stakeholder engagement facilitates the unique provision of advice, feedback and diverse perspectives. Thus far, it has helped us to understand how EVS are carried out in practice and the impact they are expected to have on care quality in primary care settings. This has aided our initial identification of appropriate documents to draw on such as localised EVS evaluations. It has also contributed to the development of our review's inclusion and exclusion criteria. As this review develops, we will engage at regular intervals with our stakeholder group to build our understanding of how mechanisms operating at the individual, group, professional and/or organisational levels produce context dependent outcomes from EVS (see also step 6). Patients are not involved in this review.

## Step 1: locating existing theories

The first step in a realist review is to undertake an initial scoping search to identify theories that begin to explain and develop our understanding of EVS. The importance of this stage is to make visible the underpinning assumptions about why certain components and processes of EVS are required, to get to the one or more desired outcomes.[29]

In the first instance, these theories will be located in the following ways: (1) iteratively drawing on exploratory searches of relevant literature and (2) consulting with key content experts who are active in the implementation or use of EVS as part of our stakeholder group engagement. Exploratory literature searches will predominantly use grey literature as a primary source of information—for example, we will focus on policy and service documents produced by NHS England and/or clinical commissioning groups on EVS. These documents will be interrogated for theories relating to the practice of EVS and their intended outcomes. This stage is not meant to be exhaustive but instead acts to provide an initial programme theory foundation. Where detail is lacking, we will endeavour to 'fill in the gaps' later on in the review.

Second, the development of a relevant programme theory will incorporate the iterative discussions within the project team. Regular meetings will be held with the aim of building, sense-making and synthesising a range of different theories into an initial programme theory. Literature, stakeholder engagement and project team discussions, along with the contents of our initial programme theory will all inform the development of an appropriate, comprehensive search strategy to be used in step 2.

## Step 2: searching for evidence

Step 2 involves one or more formal searches informed by our initial programme theory from step 1. Its goal is to identify extant literature that will be able to further inform the development of a more detailed programme theory. The process of designing, piloting and conducting the formal searches will be done with the support of an information specialist. Any modifications made to the search strategies following the pilot will be documented and implemented across source types.

The use of the following databases is anticipated: Medline, Embase, The Cochrane Library (Cochrane Database of Systematic Reviews, Cochrane Central Register of Controlled Trials, Cochrane Methodology Register) and Scopus. Any other databases identified by the information specialist as relevant will be incorporated. Forward citation searches and searching the citations contained in the reference lists of relevant documents will also be undertaken. The terminology, syntax and search structure will be informed by step 1 (ie, stakeholder collaboration, consultation with preliminary literature and initial programme theory). However, we anticipate using the following search terms for EVS within general practice: delegate*, home visit* and house calls. Subject headings relevant to each database will also be used, for example, Medical Subject Headings (MeSH) for Medline. Grey literature such as evaluations, reports,

websites, news articles and leaflets that offer useful contextual and/or conceptual information will also be used.

## Screening

All screening will be undertaken by RA. Initially, this will comprise screening of title, abstract and keywords. We will use the following inclusion criteria to determine if a document is likely to contain relevant data:

► Delegated home visiting services within general practice and its impact on individuals and/or service organisations with a healthcare need. By delegation, we are predominantly referring to the range of qualified staff able to undertake a home visit such as other GPs, advanced nurse practitioners, paramedics, nurses and emergency care practitioners.
► Document type: all study designs and documents that indicate they may contain relevant data.
► Types of participants: documents that include housebound (long term and short term) individuals with a healthcare need living within the community.
► Types of intervention: EVS, primary care visiting services, and acute home visiting services accessed via general practice within normal surgery hours (8:00–18:30).
► Outcome measures: GP workload, hospital admissions, patient health and/or satisfaction.

During the screening process, documents will be excluded if they relate to any of the following areas as these are outside the role of EVS as defined in the literature:

► Documents relating to home visiting to children as part of routine child health surveillance and maternity at home services.
► Documents relating to specialist provision of end of life/palliative care.
► Documents relating to visits provided by out of hours GP cooperatives, out of hours services, (private) social care home visits, extended hours hubs, and community-based services not accessed via general practice (eg, routine district nurse, community-based services).

A random subsample (10%) of the retrieved citations will be allocated and reviewed independently by SP to ensure consistency in the screening processes. Discussions will take place between RA and SP for any disagreements regarding the citations. For issues that cannot be resolved, the wider project team will be consulted.

## Additional searching

As the aim of the realist review is to include a broad range of documents to further inform the development of the programme theory, looking across disciplines, for example, in relation to the staffing of EVS is anticipated. Additional searches may be undertaken if there is a gap in our understanding during the refinement of the programme theory. Any additional searches that are undertaken will be discussed with the project team, in order to identify and agree on refined inclusion and exclusion criteria.

## Step 3: document selection

The selection of documents will be made in relation to their relevance (contribution to programme theory development and refinement) and rigour (credibility and trustworthiness of methods used to generate the data).[29] Documents relating to delegated home visiting services undertaken in circumstances that closely resemble the UK (ie, publicly funded healthcare setting) will be initially prioritised for inclusion and analysis. Studies from other countries with alternative healthcare structures may be drawn on later to ensure we do not miss important contributions. Using a similar criteria to Carrieri et al,[27] we define these two distinctions as having the ability to provide major and minor contributions. Document inclusion criteria for major contributions includes:

► Documents which contribute to the research questions and are conducted in the NHS.
► Documents which contribute to the research questions and are conducted in circumstances (eg, publicly funded healthcare systems) with similarities to the NHS.
► Documents which contribute to the research questions and can clearly help to identify mechanisms which could plausibly operate in the circumstances of the NHS (eg, delegated home visiting services, within hours, to patients with a healthcare need, living within the community).

## Minor contributions include

► Documents conducted in healthcare systems that are markedly different to the NHS (eg, fee-for service and private insurance scheme systems) but where the mechanisms could plausibly operate in the circumstances of the NHS.

This process, and ensuing discussions will enable reviewers to focus on data extraction and analysis of papers that provide a conceptually rich contribution while still including documents that are less conceptually rich. Decisions made regarding these classifications will be discussed by RA and SP using a random 10% selection of articles.

## Step 4: data extraction

Extraction of the data will be twofold. First, document characteristics and details will be extracted into an Excel spreadsheet with the aim of providing a descriptive overview of the documents included. Second, documents selected for inclusion will be uploaded into NVivo and coded. Details of the analytic processes may be found in step 5 (data synthesis). Data extraction will also be undertaken by RA and 10% of extracted data will be reviewed independently for consistency by another member of the team. Discussions will take place around any disagreements and extended to the project team where a resolution cannot be found. This process will be documented and the outcomes recorded.

## Step 5: data synthesis

The aim of data synthesis in realist review is to consolidate the data from previous steps to refine the initial programme theory. Data analysis and synthesis will involve the use of a realist logic analysis with the goal of using the data from the literature (ie, documents) to further develop the initial programme theory. Analysis requires interpretation and judgement of data. Data coding will be deductive (informed by our initial programme theory), inductive (come from the data within documents) and retroductive (where inferences are made based on interpretations of the data within documents about underlying causal processes—ie, mechanisms). We will use a series of questions about the relevance and rigour of content within documents as part of our process of analysis, as set out next:

► Relevance: Are sections of text within this document relevant to programme theory development?

► Rigour (judgements about trustworthiness): Are these data sufficiently trustworthy to warrant making changes to any aspect of the programme theory?

► Interpretation of meaning: If the section of text is relevant and trustworthy enough, do its contents provide data that may be interpreted as functioning as context, mechanism or outcome?

► Interpretations and judgements about context-mechanism-outcome configurations (CMOCs): What is the CMOC (partial or complete) for the data that may be interpreted as functioning as context, mechanism or outcome? Is there further data to inform this particular CMOCs contained within this document or other documents? If so, which other documents? How does this particular CMOC relate to other CMOCs that have already been developed?

► Interpretations and judgements about programme theory: How does this particular (full or partial) CMOC relate to the programme theory? Within this same document are there data which inform how the CMOC relates to the programme theory? If not, is there data in other documents? Which ones? In light of this particular CMOC and any supporting data, does the programme theory need to be changed?

Data to inform our interpretation of the relationships between contexts, mechanisms and outcomes will be sought not just within the same document, but across documents (eg, mechanisms inferred from one document could help explain the way contexts influenced outcomes in a different document). Synthesising data from different documents is often necessary to compile CMOCs, since not all parts of the configurations will always be articulated in the same document.

Within the analytic process set out above, we will use interpretive cross-case comparison to understand and explain how and why observed outcomes have occurred, for example, by comparing interventions where EVS have been 'successful' against those which have not, to understand how context has influenced reported findings. When working through the questions set out above, where appropriate we will use the following forms of reasoning to make sense of the data and refine our programme theory:

► Juxtaposition of data: for example, where data about behaviour change in one document enables insights into data about outcomes in another document.

► Reconciling of data: where data differ in apparently similar circumstances, further investigation is appropriate in order to find explanations for why these differences have occurred.

► Adjudication of data: on the basis of methodological strengths or weaknesses.

► Consolidation of data: where outcomes differ in particular contexts, an explanation can be constructed of how and why these outcomes occur differently.

## Step 6: refine programme theory

The last stage in a realist review is the refinement and testing of the programme theory.[30] In order to sense-check this, it is advisable to include the expertise of those working in practice or those who can aid in the refinement of the final theory.[31] Therefore, our final programme theory will be discussed with those undertaking EVS (eg, GPs, emergency care practitioners, nurses) and/or those involved in their organisation (GPs, receptionists). Meetings will be organised with service users and providers to discuss the findings with the goal of asking for their input to develop recommendations that are relevant to them. The active involvement of those involved in EVS is likely to improve how our findings support practice recommendations.[28] If required, the review team will revisit parts of the review that require rescrutinising. This will be undertaken until no new information is provided by the evidence or stakeholder involvement, essentially reaching theoretical saturation.[26]

This review will follow the Realist and Meta-Review Evidence Synthesis: Evolving Standards guidelines on quality and reporting.[32]

## ETHICS AND DISSEMINATION

Ensuring that the outputs of this project are useful to the construction of the best practice within general practice and commissioning services is a key priority for us. Therefore, we will produce relevant and appropriate outputs that target a range of audiences, in conjunction with stakeholder consultation:

1. Conventional academic forms. We aim to publish in a high-impact peer-reviewed journal and also present this work at academic conferences. Our hope for this is to initiate a debate about the use of EVS in primary care.

2. Plain English summaries. We aim to provide meaningful summaries of this review's findings as a method of continuous engagement with different audiences (eg, doctors, patients, commissioners and health services). We hope that this will provide an evidence-based source that can be used to inform the practice and implementation of EVS.

**Acknowledgements** The authors would like to thank our Stakeholder Group for their time and contributions to the refining of our protocol and our reviewers for their positive feedback.

**Contributors** RA, GW, KRM and SP conceptualised the study. RA designed and wrote the protocol manuscript. GW, KRM, SP, ST, A-MB and NR contributed to protocol development. GW provided methodological advice. GW and SP critically reviewed and edited the manuscript. All authors read and approved the final manuscript.

**Funding** The Evidence Synthesis Working Group is funded by the National Institute for Health Research School for Primary Care Research (NIHR SPCR) [Project Number 390].

**Disclaimer** The views expressed are those of the author(s) and not necessarily those of the NIHR, the NHS or the Department of Health.

**Competing interests** KRM is Chair, and GW is a Deputy Chair of the United Kingdom's National Institute of Health Research Health Technology Assessment Primary Care Panel.

**Patient consent** Not required.

**Ethics approval** Formal ethical approval is not required for this review as it is secondary research.

**Provenance and peer review** Not commissioned; externally peer reviewed.

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
