## [Reviewer comments · BMJ Open]

ARTICLE DETAILS

TITLE (PROVISIONAL)	Understanding the impact of delegated home visiting services accessed via general practice by community dwelling patients: A realist review protocol
AUTHORS	Abrams, Ruth; Wong, Geoffrey; Mahtani, Kamal; Tierney, Stephanie; Boylan, Anne-Marie; Roberts, Nia; Park, Sophie

VERSION 1 – REVIEW

REVIEWER	Sarah Ruiz National Institute on Disability, Independent Living, and Rehabilitation Research
REVIEW RETURNED	12-Jul-2018

GENERAL COMMENTS	This study fills an important gap in the literature by offering a realist review research protocol.
---

REVIEWER	Elizabeth Kaselitz University of Michigan, United States
REVIEW RETURNED	10-Sep-2018

GENERAL COMMENTS	This is an exceptionally thorough and well designed study protocol. I have no reservations about accepting it for publication. Two very small comments: Page 5, lines 12-16. I think this statement needs further explanation or an example on why an RCT design leads to "neglected understanding for justification and benefit". I understand the point you are making, but I think a little further explanation here is warranted. Page 6, line 15 - "indicates"
--

VERSION 1 – AUTHOR RESPONSE

Dear Dr Johnson, Dr Ruiz and Dr Kaselitz,

Thank you for your positive feedback on this protocol.

Thank you for your thoughts on the statement regarding RCTs. I have now expanded on this and it reads as follows, on page 5:

This has led to a neglected understanding of their justification and benefit,[21] particularly in relation to contextual factors (such as economic status), patient psycho-social factors (such as, for example social networks) and restricted outcome measures (e.g. mortality rates or function with a specific population).

I have also amended the spelling on Page 6. Thank you for noting this.